# Maternity and family leave experiences among female ophthalmologists in the United States

**Caroline M. Zimmermann**[1], **Courtney L. Kraus**[1], **Ashley A. Campbell**[1], **Mona A. Kaleem**[1], **Aakriti Garg Shukla**[2], **Elyse J. McGlumphy**[1]*

1 Wilmer Eye Institute, Johns Hopkins University School of Medicine, Baltimore, Maryland, United States of America, 2 Edward S. Harkness Eye Institute, Columbia University, New York, New York, United States of America

* emcglum1@jh.edu

**Data Availability Statement:** All relevant data are within the manuscript and its Supporting Information files.

## Abstract

### Objective

To evaluate family and maternity leave policies and examine the social and professional impacts on female ophthalmologists

### Participants

Participants were recruited through the Women in Ophthalmology online list-serv to complete a survey evaluating maternity leave policies and their impacts. Survey questions were repeated for each birth event after medical school for up to five birth events.

### Results

The survey was accessed 198 times, and 169 responses were unique. Most participants were practicing ophthalmologists (92%), with a minority in residency (5%), in fellowship (1.2%), on disability/leave (0.6%), or retired (0.6%). Most participants (78%) were within their first ten years of practice. Experiences were recorded for each leave event, with 169 responses for the first leave, 120 for the second, 28 for the third, and 2 for the fourth. Nearly half of participants reported the information they received about maternity leave to be somewhat or extremely inadequate (first: 50%; second: 42%; third: 41%). Many reported a greater sense of burnout after returning to work (first: 61%, second: 58%, third: 46%). A minority of participants received full pay during the first through third maternity leave events, 39%, 27%, and 33%, respectively. About a third of participants reported being somewhat or very dissatisfied with their maternity leave experience (first: 42%, second: 35%; third: 27%).

### Conclusions

Female ophthalmologists have varying experiences with maternity leave, but many encounter similar challenges. This study demonstrates that many women receive inadequate information about family leave, desire more weeks of leave, experience a wide variation in pay practices, and lack support for breastfeeding. Understanding the shared experiences of

**Funding:** The author(s) received no specific funding for this work.

**Competing interests:** The authors have declared that no competing interests exist.

women in ophthalmology identifies areas where improvements are needed in maternity leave practices within the field to create a more supportive environment for physician mothers.

## Introduction

Family and maternity leave policies play a vital role in the psychological and physical well-being of mothers and their children. Paid maternity leave has been associated with decreased rates of postpartum depression, infant mortality, and readmission to the hospital for infants and mothers as well as improved attachment of infants to mothers and increased pediatric visit attendance [1]. Paid maternity leave and duration of maternity leave have also been associated with increased duration of breastfeeding [1, 2]. Thus, in addition to addressing some of the stressors faced by female physicians, adequate maternity leave and benefits may play an important role in supporting a healthy and happy start for both mothers and infants.

Family and maternity leave policies are particularly important in ophthalmology because the percentage of female ophthalmologists is increasing. In the US, the proportion of women ophthalmologists increased from about 19% in 2009 to 27% in 2019 [3]. Despite this growth, women continue to be underrepresented among both ophthalmology residents and practicing ophthalmologists [4]. Better understanding the challenges female ophthalmologists face following birth or adoption lays a foundation for an institutional framework that more effectively supports and sustains this group. Our study aims to evaluate family and maternity leave experiences among female ophthalmologists and examine the social and psychological impacts of these practices.

## Methods

We conducted a cross-sectional study of female ophthalmologists in the United States who reported utilizing maternity or family leave during their career. The study survey was designed to assess family and maternity leave practices as well as the social and professional impact of these practices on females in ophthalmology. The Johns Hopkins Institutional Review Board approved this study, which adhered to the tenets of the Declaration of Helsinki. Informed consent was not obtained for this study as no identifiable information was collected from participants.

Participants were recruited for the study through Women in Ophthalmology online forum as well as the Ophthalmology Moms Facebook group using a voluntary and anonymous survey link that was sent to all participants of these groups. The inclusion criteria were women who were currently or previously practicing ophthalmology who reported utilizing family or maternity leave for the birth or adoption of a child in the years following graduation from medical school. The survey was accessible from May 12, 2022- June 12, 2022. Due to the overlap in membership between the two groups it is difficult to estimate the total number of individuals exposed to the survey; IP addresses were used to ensure unique responses.

The survey was created and administered using Qualtrics © software (Qualtrics, Provo, UT). The survey design was based off a previously published study on family leave in physician mothers by Juengst et al. [5]. The survey in the Juengst et al. paper was developed using a modified Delphi process with a panel of experts that included physicians from multiple specialties, a nurse, a policy expert, and a lawyer. Our survey contained a set of initial questions regarding current professional position, history of practice, number of birth or adoption events, and

impact of career on family planning. It included 30 questions concerning each birth or adoption event after medical school, for up to five events, including family and maternity leave benefits, financial impacts, breastfeeding, social support, and satisfaction. Survey questions were structured as yes/no, multiple choice, or short answer. A final free response question was included for participants to share any other comments about their family and maternity leave experiences. The survey is provided in the S1 Fig.

Descriptive statistics, including frequencies, percentages, means, standard deviations, and ranges, were used to characterize the study participants and their experiences with family and maternity leave. Stata (StataCorp. 2021. *Stata Statistical Software*: *Release 17*. College Station, TX: StataCorp LLC) was utilized for data analysis.

## Results

### Recruitment

The survey link was accessed a total of 198 times; 5 individuals were excluded by failing to meet inclusion criteria. In the remaining 193 surveys, 12 surveys were excluded due to the inability to verify a unique participant through a duplicate IP address. Lastly, 12 surveys were excluded due to incomplete data in which participants failed to answer any questions after the screening questions. A flowchart of recruitment can be visualized in Fig 1.

### Participants

Details on participants are presented in Table 1. The majority of participants were currently practicing (156/169 [92%]), with a minority in ophthalmology residency (9/169 [5.3%]), ophthalmology subspecialty fellowship (2/169 [1.2%]), ophthalmologists currently on disability/leave (1/169 [0.6%]), and retired ophthalmologists (1/169 [0.6%]). Participants averaged 8.2 years (SD 6.2, Range 0–36) in practice. Most participants (123/157 [78%]) were within their first ten years of practice. Experiences were recorded for each leave event, with 169 responses for the first leave, 120 for the second, 28 for the third, and 2 or the fourth. No participants reported more than 4 leave events. No adoptions were reported. Over half of participants reported that someone recommended they delay family planning at some point in their career (95/169 [56%]). A majority of participants felt that their career probably or definitely impacted their family planning (141/169 [83%]). Detailed characteristics can be found in Table 1.

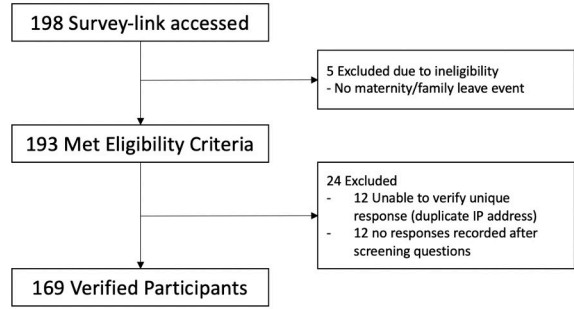

**Fig 1. Flowchart showing included and excluded survey participants and reasons for exclusion.**

**Table 1. Baseline characteristics of survey participants.**

| | | N (%) |
|---|---|---|
| **No.** | | **169** |
| Current Position | Ophthalmology Resident | 9 (5) |
| | Ophthalmology Fellow | 2 (1) |
| | Practicing Ophthalmologist | 156 (92) |
| | Ophthalmologist on Leave/Disability | 1 (0.6) |
| | Retired Ophthalmologist | 1 (0.6) |
| | Other | - |
| Years in Practice | Mean (SD, Range) | 8.2 (6.2, 0–36) |
| | 0–5 | 62 (0.39) |
| | 6–10 | 61 (0.39) |
| | 11–15 | 19 (0.12) |
| | 16–20 | 8 (0.05) |
| | >20 | 7 (0.04) |
| Mean Hours Worked per Week | <20 | 1 (0.6) |
| | 20–40 | 68 (40) |
| | 41–60 | 87 (51) |
| | 61–80 | 12 (7) |
| | >80 | 1 (0.6) |
| Number of Maternity/Family Leave Events | 1 | 49 |
| | 2 | 92 |
| | 3 | 26 |
| | 4 | 2 |
| | 5 | 0 |
| Current Specialty | Resident/fellow | 10 (6) |
| | Comprehensive | 39 (23) |
| | Glaucoma | 31 (18) |
| | Cornea | 22 (13) |
| | Retina | 19 (11) |
| | Medical Retina | 5 (3) |
| | Pediatrics | 19 (11) |
| | Neuro-ophthalmology | 1 (0.6) |
| | Oculoplastics | 13 (8) |
| | Uveitis | 4 (2) |
| | Multiple | 5 (3) |
| | Other | 2 (1) |

## Family leave and return to work experiences

Characterization of family and maternity leave policies and pay are classified by birth event in Table 2. Less than half of women reported that their workplace had a maternity leave policy (first: 67/158 [42%]; second: 49/106 [46%]; third: 9/22 [41%]; fourth: 0/2 [0%]), and nearly half of participants found the information they received about maternity leave to be somewhat or extremely inadequate (first: 79/159 [50%]; second: 45/107 [42%]; third: 9/22 [41%]). Across all birth events, the average number of weeks desired for leave was 12–13 weeks (average [range] first: 13.2 [3–52], second: 12.7 [4–28], third: 12 [3–20], fourth: 15), which was greater than the average number of weeks taken for all leave events (first: 8.6 [0–26], second: 9.9 [2–28], third: 11.1 [3–40], and fourth: 6). Maternity/family leave days came from a variety of sources,

primarily vacation days (first: 92/169 [54%], second: 47/105 [45%], third: 11/22 [50%]) and sick leave (first: 58/169 [34%], second: 36/105 [34%], third: 6/22 [27%]). A minority of participants received full pay during the first through third maternity leave events, 39%, 27%, and 33%, respectively. However, most reported no financial hardship during this time (first: 63/93 [68%], second: 23/44 [52%], third: 5/6 [83%]). When applicable, a majority of participants did report a sizeable reduction in their clinical production bonus with first and second leave (first: 58/78 [74%], second: 59/77 [77%], third: 4/19 [21%]). Additional details can be found in Table 2.

Family and maternity leave support experiences are detailed in Table 3. Participants experienced pressure to answer emails and calls (first: 28%, second: 26%, third: 26%), continued academic expectations (first: 29%, second: 19%, third: 19%), and pressure to return to work early (first: 30%, second: 18%, third: 22%). Less frequently but still notably, participants experienced derogatory comments by colleagues and staff (first: 23%, second: 10%, third: 26%), financial pressures from coworkers or their institution (first: 19%, second: 18%, third: 22%), and anger from patients (first: 14%, second: 13%, third: 15%). Nearly half of participants had partners who took leave (first: 73/144 [51%], second: 45/96 [47%], third: 11/20 [55%]). Most participants felt supported by their colleagues and administration during leaves and upon return to work (first:116/156 [74%], second: 84/104 [81%], third: 15/22 [68%]). A majority experienced a greater sense of burnout with return to work for the first and second leave experiences (first: 95/156 [61%], second: 60/103 [58%], third:10/22 [46%]). Post-partum depression, anxiety, obsessive compulsive disorder, or other mental health disorders in the post-partum period were experienced by over a quarter of women during their first leave experience (first: 44/156 [28%], second: 13/103 [13%], third: 1/22 [4.5%]). Overall, about a third of participants reported being somewhat or very dissatisfied with their maternity leave experience (first: 66/156 [42%], second: 36/103 [35%]; third: 6/22 [27%]).

Most participants reported breastfeeding (first: 139/157 [89%], second: 96/106 [91%], third: 20/22 [91%]; Table 4). Of those who breastfed, over a quarter reported their employer provided slightly, moderately, or extremely inadequate time for lactation support (first: 48/131 [37%], second: 22/87 [25%], third: 8/20 [40%]) and a smaller percentage were made to feel guilty for time missed during lactation support (first: 37/130 [29%], second: 16/89 [18%], third: 3/20 [15%]).

## Discussion

Our study examines family leave experiences among female ophthalmologists and the impact on their personal and professional lives. Nearly 80% of study participants were in their first ten years of practice and represented currently practicing physician mothers in the United States of America. Half of participants reported receiving inadequate information about leave and no or partial pay during leave. Although they felt supported by their colleagues and administration, the majority experienced a greater sense of burnout upon returning to work. Our findings identify areas of improvement in family leave practices and suggest that female ophthalmologists who experience motherhood have shared challenges.

Among study participants, over half did not have a maternity leave policy established in the workplace, and close to half found information regarding family and maternity leave to be inadequate. Lack of clarity surrounding maternity leave policies is also found across other medical specialties [5, 6]. As some study participants commented, navigating maternity leave can be incredibly difficult, especially when one is the first in the workplace to take maternity leave. On average, participants took more leave than was available using maternity leave days. However, this amount of time was still less than what participants desired to take, a finding

**Table 2. Family and maternity leave policies and pay.**

| | | First | Second | Third |
|---|---|---|---|---|
| N | | 165 | 107 | 21 |
| Age Y (SD, Range) | | 32.6 (3.0, 26–44) | 34.8 (2.7, 29–44) | 37 (1.7, 34–42) |
| Pregnancy without intervention | | 133 | 94 | 19 |
| Pregnancy with intervention | | 32 | 13 | 2 |
| If applicable, support for fertility appointments and procedures | Extremely inadequate | 9 (30) | 2 (15) | 1 (50) |
| | Somewhat inadequate | 3 (10) | 1 (8) | 0 |
| | Neutral | 8 (27) | 8 (62) | 1 (50) |
| | Mostly adequate | 9 (30) | 2 (15) | 0 |
| | Extremely adequate | 1 (3) | 0 | 0 |
| Adoption | | 0 | 0 | 0 |
| Practice Setting | Resident/Fellow | 39 (25) | 7 (7) | 0 |
| | Solo practice | 16 (10) | 4 (4) | 0 |
| | PP ≤5 physicians | 3 (2) | 5 (5) | 1 (5) |
| | PP >5 physicians | 36 (23) | 36 (34) | 7 (32) |
| | Hospital based practice | 17 (11) | 13 (12) | 3 (14) |
| | Academic practice | 10 (6) | 8 (8) | 2 (9) |
| | Private equity group | 37 (23) | 32 (30) | 9 (41) |
| Did your workplace have a maternity leave policy? | Yes | 67 (42) | 49 (46) | 9 (41) |
| | No | 79 (50) | 52 (49) | 13 (59) |
| | Unsure | 12 (8) | 5 (5) | 0 |
| Provided with adequate information about family/maternity leave and was it easy to understand? | Extremely inadequate | 48 (30) | 25 (23) | 6 (27) |
| | Somewhat inadequate | 31 (19) | 20 (19) | 3 (14) |
| | Neutral | 24 (15) | 19 (18) | 4 (18) |
| | Somewhat adequate | 33 (21) | 20 (19) | 4 (18) |
| | Extremely adequate | 23 (14) | 23 (21) | 5 (23) |
| Weeks Mean (SD) (Range) | | | | |
| | Weeks of leave available | 7.28 (4.5) (0–24) | 8.57 (5.0) (0–24) | 10.6 (6.2) (0–24) |
| | Weeks of leave taken | 8.6 (4.0) (0–26) | 9.9 (4.2) (2–28) | 11.1 (7.6) (3–40) |
| | Weeks of leave desired | 13.2 (5.6) (3–52) | 12.7 (4.3) (4–28) | 12 (3.4) (3–20) |
| Source of leave | | | | |
| | Sick leave | 58 (34) | 36 (34) | 6 (27) |
| | Vacation | 92 (54) | 47 (45) | 11 (50) |
| | Paid time off | 35 (21) | 24 (23) | 6 (27) |
| | Specified maternity leave | 52 (31) | 32 (30) | 8 (36) |
| | Other | 57 (34) | 42 (40) | 6 (27) |
| Pay | No pay | 43 (27) | 27 (43) | 2 (22) |
| | Partial pay | 37 (23) | 17 (27) | 4 (44) |
| | Full pay | 62 (39) | 17 (27) | 3 (33) |
| | Unable to recall/other | 18 (11) | 2 (3) | - |
| Financial hardship during leave | No | 63 (68) | 23 (52) | 5 (83) |
| | Yes | 27 (29) | 21 (48) | 1 (17) |
| Sizable reduction in clinical production bonus | No | 20 (26) | 18 (23) | 4 (21) |
| | Yes | 58 (74) | 59 (77) | 15 (79) |

*(Continued)*

**Table 2.** (Continued)

| | | First | Second | Third |
|---|---|---|---|---|
| Subsequent reduction in work hours upon return | No | 112 (72) | 84 (79) | 20 (95) |
| | Yes | 23 (14) | 19 (9) | 1 (5) |
| | NA | 22 (14) | 12 (11) | 0 |

which mirrors the experience of physician mothers across specialties [5]. Participants took their leave from multiple sources, including specified maternity leave days, sick leave, vacation, paid time off, and unpaid time off, which further demonstrates that specified maternity leave days alone were inadequate. Notably, over half of participants utilized vacation days for maternity leave. Vacation days are meant for rest and recuperation from work, but time spent on maternity leave is often far from restful as mothers face the physical and emotional demands of caring for a newborn, and if applicable, the physical recovery from the birthing event. These findings suggest that providing more information regarding leave policies and building in additional time for maternity leave may help female ophthalmologists feel more supported during and after pregnancy, which may in turn improve experiences of burnout.

Some participants noted financial struggles during leave, with nearly a third of women experiencing financial hardships after the birth of their first child and nearly three quarters of women experiencing a sizeable loss in clinical production bonus after the birth of their first and second children. Lost income during leave may also not fully capture the economic impact of family planning. Some may also have to curtail their clinical volume to allow time for lactation support upon return to work. Twenty-seven percent of participants reported no paid leave while 23% reported partial paid leave after the birth of their first child. Furthermore, several participants commented that even after their return from leave, they struggled for months to rebuild their clinical volume. Female ophthalmologists already face financial disparities as their median reimbursements remain less than that for males [7]. Additional financial stressors from maternity leave only add to the inequities for women in the field. In particular, childcare can be a substantial cost for new parents, and over 70% of participants reported having a nanny or using daycare as the primary means of childcare for the first year of life. Participants also felt professional stress while on leave, noting most frequently the pressure to keep up with academic expectations, answer emails and calls, and even return to work early. Some participants experienced derogatory comments from colleagues and administration which is particularly troublesome in this time of vulnerability. While it is difficult to lessen the financial burden of parental leave on a small physician group, large physician groups, private equity firms, and academic institutions should establish a robust system capable of accommodating periods of leave for medical and family planning purposes. Furthermore, due to the demanding nature of many physician roles, individuals should be given an appropriate length of time for leave without undue incentives for early return to encourage proper family bonding, maternal mental and physical health, and the establishment of lactation if applicable.

Return to work, while challenging alone, is further complicated for those who choose to breastfeed beyond leave. Most participants wished or chose to breastfeed. About half of those women felt they received adequate time for lactation support, reported they were not made to feel guilty for the time spent during lactation, and did not feel like their job negatively impacted breastfeeding, while the other half of women expressed more neutral or negative experiences with breastfeeding upon return to work. These data show that women in ophthalmology have varying experiences with breastfeeding and could benefit from standardized protocols to protect time and space for breastfeeding during the workday. Since lactation support

**Table 3. Family and maternity leave support experiences.**

| | | First | Second | Third |
|---|---|---|---|---|
| Did you feel supported by colleagues and admin during leave and upon return to work? | Definitely yes | 57 (37) | 50 (48) | 9 (41) |
| | Probably yes | 59 (38) | 34 (33) | 6 (27) |
| | Probably not | 24 (15) | 12 (12) | 4 (18) |
| | Definitely not | 16 (10) | 8 (8) | 3 (14) |
| Did you consider leaving your job after the birth of your child? | Yes | 35 (22) | 16 (15) | 4 (18) |
| | No | 116 (74) | 84 (81) | 17 (77) |
| | I did leave | 4 (3) | 4 (4) | 1 (5) |
| | Unable to recall | 1 (0.6) | 0 | 0 |
| Did you suffer with post-partum depression, anxiety, obsessive compulsive disorder, or other mental health disorders in the post-partum period? | Yes | 44 (28) | 13 (13) | 1 (5) |
| | Maybe | 19 (12) | 10 (1) | 3 (14) |
| | No | 93 (60) | 80 (78) | 18 (82) |
| Childcare | Nanny/Home | 74 (47) | 57 (55) | 14 (64) |
| | Relative/Family | 32 (21) | 7 (7) | 1 (5) |
| | Daycare | 36 (23) | 28 (27) | 5 (23) |
| | Stay at home spouse | 8 (5) | 8 (8) | 0 |
| | I stayed at home | 2 (1) | 1 (1) | 1 (5) |
| | Other | 4 (3) | 3 (3) | 1 (5) |
| Did you feel a greater sense of burnout with return to work? | Yes | 95 (61) | 60 (58) | 10 (45) |
| | Maybe | 30 (19) | 14 (14) | 3 (14) |
| | No | 31 (20) | 29 (28) | 9 (41) |
| Participants reported experiencing (%): | | | | |
| | Derogatory comments by colleagues or staff | 23 | 10 | 26 |
| | Financial pressure from coworkers/institution | 19 | 18 | 22 |
| | Termination from position | 4 | 3 | 4 |
| | Delay in partnership | 8 | 8 | 7 |
| | Anger from patients for rescheduled visits etc | 14 | 13 | 15 |
| | Continued pressure to answer emails/calls | 28 | 26 | 26 |
| | Continued expectation to participate in academic endeavors | 29 | 19 | 19 |
| | Pressure to return early | 30 | 18 | 22 |
| Did circumstances impact future family planning | Definitely yes | 33 (21) | 24 (23) | 7 (32) |
| | Probably yes | 26 (17) | 11 (11) | 0 |
| | Might or might not | 21 (13) | 8 (8) | 1 (5) |
| | Probably not | 42 (27) | 29 (28) | 7 (32) |
| | Definitely not | 34 (22) | 31 (30) | 7 (32) |
| Did your partner take leave | Yes | 73 (51) | 45 (47) | 11 (55) |
| | No | 71 (49) | 51 (53) | 9 (45) |
| Overall level of satisfaction | Very unsatisfied | 37 (24) | 17 (17) | 3 (14) |
| | Somewhat unsatisfied | 29 (19) | 19 (18) | 3 (14) |
| | Neither satisfied nor unsatisfied | 18 (12) | 12 (12) | 4 (18) |
| | Somewhat satisfied | 45 (29) | 33 (32) | 4 (18) |
| | Very satisfied | 27 (17) | 22 (21) | 8 (36) |

is a legal requirement and breastfeeding has many positive health benefits for both mother and baby, support and accommodations for lactation support should be a priority [8]. Individuals with a history of lactation have been shown to have a reduced incidence of diabetes mellitus type 2, breast cancer, and ovarian cancer. Not breastfeeding or early cessation of breastfeeding is associated with an increased risk of postpartum depression [9].

**Table 4. Breastfeeding experiences reported during family and maternity leave events.**

| | | First | Second | Third |
|---|---|---|---|---|
| Breastfeeding | Yes | 139 (89) | 96 (91) | 20 (91) |
| | Unable | 13 (8) | 7 (7) | 0 |
| | No | 5 (3) | 3 (3) | 2 (9) |
| Did your employer provide adequate time for lactation support? | Extremely inadequate | 16 (12) | 6 (7) | 3 (15) |
| | Moderately inadequate | 22 (17) | 10 (11) | 4 (2) |
| | Slightly inadequate | 10 (8) | 6 (7) | 1 (5) |
| | Neutral | 11 (8) | 8 (9) | 4 (20) |
| | Slightly adequate | 8 (6) | 11 (13) | 1 (5) |
| | Moderately adequate | 37 (28) | 21 (24) | 3 (15) |
| | Extremely adequate | 27 (21) | 25 (29) | 4 (20) |
| Were you made to feel guilty for time missed during lactation? | Yes | 37 (28) | 16 (18) | 3 (15) |
| | Maybe | 18 (14) | 18 (20) | 5 (25) |
| | No | 75 (58) | 55 (62) | 12 (60) |
| Do you feel your job impacted breastfeeding in a negative way? | Yes | 62 (47) | 24 (28) | 7 (35) |
| | Maybe | 19 (14) | 21 (24) | 3 (15) |
| | No | 52 (39) | 42 (48) | 10 (50) |

Many participants experienced postpartum mental health difficulties and even more reported a greater sense of burnout upon returning from their leave. The rate of postpartum depression or other mental health issues among female ophthalmologists in our study was nearly 30% for the first birth experience, which is over double the national prevalence of postpartum depression [10]. Burnout is highly prevalent among women in medicine, and the additional challenge of navigating maternity leave and its associated personal, professional, and financial facets may contribute further to this problem [11, 12]. Although female ophthalmologists are underrepresented in practice, the number of women entering the field continues to grow [3, 4]. In order to support this growth and protect against increased burnout, we must be attentive to the challenges female ophthalmologists face, especially surrounding maternity leave, and work toward creative solutions to more effectively support and empower women during and after the birth or adoption of a child.

Despite many reported difficulties, many participants reported feeling supported by colleagues and administration both during leave and upon return and about half were somewhat or very satisfied with their maternity leave experience overall.

This study has several potential limitations. This survey represents the experiences of 169 female ophthalmologists and ophthalmologists in training, a small fraction of the eligible candidates. Our study asked individuals to recollect their prior maternity/family leave experiences from memory introducing the possibility for recall bias; individuals may fail to remember previous events or experiences accurately. Fortunately, most participants were in the first ten years of practice, and one-third in the first 0–5 years of practice, representing those with recent parental leave experiences. Furthermore, given most participants were within the first ten years of their career, this study may not accurately represent practices and experiences of late-career physicians. It is also possible that individuals who had prior experiences with maternity/family leave that were undesirable may be more likely to take the survey; while many women experienced difficulties during leave, there were also many participants who reported a positive experience. Our survey also only addresses the experiences of female ophthalmologists; male ophthalmologists are also subject to similar stressors in family planning and leave, and additional studies are warranted to further characterize their experience. Additionally,

future studies could compare the experience of physician ophthalmologists to other instructors in the ophthalmology department, including Ph.D.s and postdoctoral scholars.

Overall, this study brings to light the family and maternity leave experiences of female ophthalmologists. As the female representation grows in medicine and ophthalmology, the policies and practices around maternal health and benefits need to reflect the changing demographic. It is our hope that highlighting the experiences of women across our nation will serve as a foundation and impetus for standardizing and implementing family and maternal leave policies which promote infant bonding, lactation, and maternal physical and mental well-being.

## Supporting information

**S1 Fig. Maternity leave in ophthalmology survey.** This is a copy of the survey administered to participants in the study.
(DOCX)

## Author Contributions

**Conceptualization:** Elyse J. McGlumphy.

**Data curation:** Elyse J. McGlumphy.

**Formal analysis:** Caroline M. Zimmermann, Elyse J. McGlumphy.

**Investigation:** Elyse J. McGlumphy.

**Methodology:** Elyse J. McGlumphy.

**Project administration:** Elyse J. McGlumphy.

**Supervision:** Elyse J. McGlumphy.

**Writing – original draft:** Caroline M. Zimmermann, Elyse J. McGlumphy.

**Writing – review & editing:** Caroline M. Zimmermann, Courtney L. Kraus, Ashley A. Campbell, Mona A. Kaleem, Aakriti Garg Shukla, Elyse J. McGlumphy.

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
