## [Decision Letter · Decision Letter 0]

20 Dec 2022

PONE-D-22-29362Maternity and family leave experiences among female ophthalmologistsPLOS ONE

Dear Dr. Mcglumphy,

Thank you for submitting your manuscript to PLOS ONE. After careful consideration, we feel that it has merit but does not fully meet PLOS ONE’s publication criteria as it currently stands. Therefore, we invite you to submit a revised version of the manuscript that addresses the points raised during the review process.

We look forward to receiving your revised manuscript.

Kind regards,

Isabelle Jalbert, PhD

Academic Editor

PLOS ONE

Journal Requirements:

Additional Editor Comments:

Congratulations for a very interesting study. Your reviewers have asked for several modifications and clarification which I encourage you to carefully consider. In addition, I support the requests to:

- include "in the United States" in the title of the manuscript

- provide a copy of the questionnaire as a Table or as Supplementary material

Please also consider the following:

- provide more details on how the survey questions were developed: were they based on existing instruments? were these pilot test? on who? etc.

- provide an estimate of the response rate based on the maximum number of people the survey was sent to

- comment on the generalizability of the results to the population of female ophthalmologists in the USA based on the data you have collected and what is known about the general characteristics of this population

Reviewers' comments:

Reviewer's Responses to Questions

**Comments to the Author**

1. Is the manuscript technically sound, and do the data support the conclusions?

Reviewer #1: Yes

Reviewer #2: Yes

2. Has the statistical analysis been performed appropriately and rigorously? 

Reviewer #1: N/A

Reviewer #2: No

3. Have the authors made all data underlying the findings in their manuscript fully available?

Reviewer #1: Yes

Reviewer #2: Yes

4. Is the manuscript presented in an intelligible fashion and written in standard English?

Reviewer #1: Yes

Reviewer #2: Yes

5. Review Comments to the Author

Reviewer #1: It is interesting to see the real data provided by female ophgthalmologists regarding their experience with maternity leave and time for breast feeding and lactation support. Very useful paper, well done.

The authors should refrain from inserting their personal opinion and stay with the facts.

Minor comments

The first paragraph of the introduction is biased and removes the interest in reading this paper as the conclusions are already drawn. I suggest to remove line 1-10 from the introduction and discuss this in the discussion.

Please remove decimals if percentage is 10 or higher (not 40.1 but 40 %)

Page 3 line 19: if it increases from 1 to 2 %, this would be a moot paper. Please indicate what the percentage was 10 and 5 years ago and now.

This paper is only about the United States of America, regrettably often a backwater if looking at maternity leave. Please change the title of the paper to indicate that this study took place in the United States of America.

Page 4; which ophthalmology mothers group? Can you specify how this group was contacted

Table 1: mean hours worked: you mean per week? Please add.

N (%): 9 (0.05): the 0.05 is not percentage. Please change to 5%. Same below. Please change in all tables where you mean percentage.

Table 3: what do the numbers in the section: Participants reported experiencing… stand for (0.23 etc)?

Discussion: Please refrain from stating you are the first. Line 128: the question was not only about burnout, so this sentence should be modified. Same applies to line 212.

No information was provided about male dominance at the work place. Please adjust: “As some study participants commented, navigating maternity leave can134 be incredibly difficult, especially when one is working in a male-dominated workplace or is the first in the workplace to take maternity leave.”

Line 149: Most did not report financial hardship. Line is biased.

Why do you not call Express Milk: lactation support? Is it really called Express Milk time?

Line 198: add : in the United States of America.

an interesting paper about career choices and satisfaction early on in the USA is:

Factors Influencing Career Decisions and Satisfaction Among Newly Practicing Ophthalmologists.

Gedde SJ, Feuer WJ, Crane AM, Shi W. Am J Ophthalmol. 2022 Feb;234:285-326. doi: 10.1016/j.ajo.2021.06.011. Epub 2021 Jun 23. PMID: 34157277

Reviewer #2: The study evaluates the professional and social effect of maternity and family leave policies on female clinicians in the field of ophthalmology. It demonstrates the challenges currently faced by the female ophthalmologists which could provide evidence to improve the current system of family leaves to help and support the physician mothers and prevent burnout. I believe the manuscript provides important information and evidence to form basis for an improvement in the current family and maternity leave policies and thus, I recommend it to be accepted with corrections.

1. Methodology mentions the use of voluntary and anonymous survey link for recruiting the participants. Please mention the name and provide links for theses surveys in the methodology section.

2. Did the study include transgender population that identify themselves as women?

3. A copy of the survey form should be added in the methodology as a figure, or in the supplementary methodology section, for the ease of the readers.

4. Please describe the type of statistics used, number of participates, Mean, SD calculations, software used etc in the methodology section.

5. The study could also include differences in experiences of physicians with/without interventions during their pregnancy.

6. The study could also include scope of including researchers such as Ph.D.’s , Postdoctoral scholars, instructors in the ophthalmology department of the hospital set up in the future, and if their experiences align with the physician ophthalmologists.

7. Did the study include workplace (hospital) daycares for the physicians and if there were any financial support for the daycare cost? Daycare cost is another essential factor for new mothers undergoing financial stress and should be discussed in the discussion section.

Minor typos

1. Table 2 require formatting, there are some extra lines.

2. There are different font sizes in table 2 which should be consistent

3. The labelling of the tables are different, please current the headings “label Leave 1, leave 2 and leave 3” in Table 2 to first, second and third as mentioned in Table 3 and 4 to maintain consistently in the tables for the ease of readers.

4. In some places the numbers are written alphabetically while in other places its written numerically kindly correct it. For eg, in line 154 and 155.

5. Figure one is blurred and needs to be in a high resolution.

6. PLOS authors have the option to publish the peer review history of their article (what does this mean?). If published, this will include your full peer review and any attached files.

Reviewer #1: No

Reviewer #2: No

---

## [Author Response · Author response to Decision Letter 0]

23 Jan 2023

January 17, 2023

Dear Editor, 

Please find our updated manuscript, Maternity and Family Leave Experiences Among Female Ophthalmologists in the United States, for consideration for publication in PLOS ONE. Here, we have surveyed female ophthalmologists who have utilized maternity leave for the birth of a child following medical school to evaluate maternity leave policies and examine the social and professional impacts on females in ophthalmology. We believe our findings identify potential areas of improvement in maternity and family leave practices within the field and will help create a more supportive environment for mothers in ophthalmology. We appreciate the thoughtful comments from the reviewers and have included our responses to their recommendations below.

Additional Editor Comments:

In addition, I support the requests to:

- include "in the United States" in the title of the manuscript

• The title of the manuscript has been adjusted to include the phrase “in the United States.”

- provide a copy of the questionnaire as a Table or as Supplementary material

• The questionnaire is now available as a supplementary material.

Please also consider the following:

- provide more details on how the survey questions were developed: were they based on existing instruments? were these pilot test? on who? etc.

• We have updated our methods section to explain that our survey questions were based off a study published in JAMA that evaluated family/maternity leave experiences for physician mothers across specialties. This survey was created using a modified Delphi process with a panel of experts that included physicians from multiple specialties, a nurse, a policy expert, and a lawyer.

- provide an estimate of the response rate based on the maximum number of people the survey was sent to

• There are approximately 1,700 women in the Ophthalmology Mom Facebook group and the number of women in Women in Ophthalmology is not publicly available. There is also considerable overlap between the groups, many women in the Ophthalmology Mom Facebook group are also members of Women in Ophthalmology therefore making it difficult to estimate the response rate. The survey was made available via a link to both of these populations and thus it is unclear how many individuals saw or had access to the link at the time of publication of the survey. 

- comment on the generalizability of the results to the population of female ophthalmologists in the USA based on the data you have collected and what is known about the general characteristics of this population

• As mentioned in our paper, this data mostly represents women who have taken leave within the last 10 years, as most participants have stated such. Many participants are currently practicing ophthalmologists. We believe this is a diverse group of participants when it comes to practice setting with a mix of trainees and those from private and academic settings. 

Review Comments to the Author

Reviewer #1: 

Minor comments

The first paragraph of the introduction is biased and removes the interest in reading this paper as the conclusions are already drawn. I suggest to remove line 1-10 from the introduction and discuss this in the discussion.

• We have removed the first paragraph from the introduction. Many of the same ideas are examined in the discussion.

Please remove decimals if percentage is 10 or higher (not 40.1 but 40 %)

• All percentages have been corrected so there are no decimals in percentages that are 10 or higher.

Page 3 line 19: if it increases from 1 to 2 %, this would be a moot paper. Please indicate what the percentage was 10 and 5 years ago and now.

• We provided the percentage increase from 2009 (19%) to 2019 (27%).

This paper is only about the United States of America, regrettably often a backwater if looking at maternity leave. Please change the title of the paper to indicate that this study took place in the United States of America.

• The title has been adjusted to reflect that this study took place in the United States.

Page 4; which ophthalmology mothers group? Can you specify how this group was contacted

• We have updated the methods to specify the Ophthalmology Moms Facebook group that was utilized. An anonymous survey link was posted to the group’s Facebook page.

Table 1: mean hours worked: you mean per week? Please add.

• Table 1 was updated to specify mean hours worked per week.

N (%): 9 (0.05): the 0.05 is not percentage. Please change to 5%. Same below. Please change in all tables where you mean percentage.

• The percentages were corrected in all the tables.

Table 3: what do the numbers in the section: Participants reported experiencing… stand for (0.23 etc)?

• These numbers stand for the percentage of participants who reported experiencing the different situations.

Discussion: Please refrain from stating you are the first. 

• We have deleted this statement from the discussion.

Line 128: the question was not only about burnout, so this sentence should be modified. 

• The question was about burnout upon return to work, so the sentence reflects the survey question.

Same applies to line 212.

• This line was modified to remove the phrase “is the first to.”

No information was provided about male dominance at the work place. Please adjust: “As some study participants commented, navigating maternity leave can134 be incredibly difficult, especially when one is working in a male-dominated workplace or is the first in the workplace to take maternity leave.”

• This line was modified to remove the phrase about the male-dominated workplace.

Line 149: Most did not report financial hardship. Line is biased.

• This line was modified to reflect that financial hardship was a problem faced by only some of participants.

Why do you not call Express Milk: lactation support? Is it really called Express Milk time?

• All phrases with milk expression have been updated to say lactation support.

Line 198: add : in the United States of America.

• This phrase has been added.

Reviewer #2: 

1. Methodology mentions the use of voluntary and anonymous survey link for recruiting the participants. Please mention the name and provide links for theses surveys in the methodology section.

• The methods section was updated to include the name of the group that received the survey link. Additionally, the survey is now included as a supplemental material.

2. Did the study include transgender population that identify themselves as women?

• This study did not have participants indicate specific gender identifications. Rather, any ophthalmologist who identified as a mother and experienced family/maternity leave was invited to participate. Our survey also included neutral language in our survey (such as chest-feed) as to not exclude individuals from the transgender population.

3. A copy of the survey form should be added in the methodology as a figure, or in the supplementary methodology section, for the ease of the readers.

• The survey will be included as a supplemental figure for readers to access.

4. Please describe the type of statistics used, number of participates, Mean, SD calculations, software used etc in the methodology section.

• We have updated the methods section to include more information on statistics. The number of participants is included in the results section. 

5. The study could also include differences in experiences of physicians with/without interventions during their pregnancy.

• While we agree with this, we wish to pursue this for additional subsequent papers. 

6. The study could also include scope of including researchers such as Ph.D.’s , Postdoctoral scholars, instructors in the ophthalmology department of the hospital set up in the future, and if their experiences align with the physician ophthalmologists.

• This would be an interesting comparison to make in a future paper. In the discussion, we have included the recommendation for future research to evaluate the experiences of other instructors in the ophthalmology department.

7. Did the study include workplace (hospital) daycares for the physicians and if there were any financial support for the daycare cost? Daycare cost is another essential factor for new mothers undergoing financial stress and should be discussed in the discussion section.

• Because the study was looking at ophthalmologists at different stages of training and in different professional settings, we did not ask about workplace daycares or financial support for daycares. We do agree that daycare cost is a significant financial stressor for new mothers, and about a quarter of survey participants reported using daycare for each child. The cost of childcare was added to the discussion as a significant financial stressor.

Minor typos

1. Table 2 require formatting, there are some extra lines.

• The extra lines have been removed.

2. There are different font sizes in table 2 which should be consistent

• The font sizes have been adjusted to be consistent.

3. The labelling of the tables are different, please current the headings “label Leave 1, leave 2 and leave 3” in Table 2 to first, second and third as mentioned in Table 3 and 4 to maintain consistently in the tables for the ease of readers.

• The labeling was updated on Table 2 to match the labeling in Table 3 and 4.

4. In some places the numbers are written alphabetically while in other places its written numerically kindly correct it. For eg, in line 154 and 155.

• Numbers are written alphabetically only when the first word of a sentence.

5. Figure one is blurred and needs to be in a high resolution.

• Figure one has been updated to a higher resolution.

---

## [Decision Letter · Decision Letter 1]

3 Apr 2023

Maternity and family leave experiences among female ophthalmologists in the United States

PONE-D-22-29362R1

Dear Dr. Mcglumphy,

We’re pleased to inform you that your manuscript has been judged scientifically suitable for publication and will be formally accepted for publication once it meets all outstanding technical requirements.

Kind regards,

Bidisha Banerjee, Ph.D.

Academic Editor

PLOS ONE

Additional Editor Comments (optional):

Reviewers' comments:

Reviewer's Responses to Questions

**Comments to the Author**

1. If the authors have adequately addressed your comments raised in a previous round of review and you feel that this manuscript is now acceptable for publication, you may indicate that here to bypass the “Comments to the Author” section, enter your conflict of interest statement in the “Confidential to Editor” section, and submit your "Accept" recommendation.

Reviewer #1: All comments have been addressed

Reviewer #2: All comments have been addressed

2. Is the manuscript technically sound, and do the data support the conclusions?

Reviewer #1: Yes

Reviewer #2: Yes

3. Has the statistical analysis been performed appropriately and rigorously? 

Reviewer #1: Yes

Reviewer #2: Yes

4. Have the authors made all data underlying the findings in their manuscript fully available?

Reviewer #1: Yes

Reviewer #2: Yes

5. Is the manuscript presented in an intelligible fashion and written in standard English?

Reviewer #1: Yes

Reviewer #2: Yes

6. Review Comments to the Author

Reviewer #1: (No Response)

Reviewer #2: The authors have addressed all the comments in an informative manner, and I recommend the paper to be accepted

7. PLOS authors have the option to publish the peer review history of their article (what does this mean?). If published, this will include your full peer review and any attached files.

Reviewer #1: No

Reviewer #2: No

---

## [Editor Report · Acceptance letter]

14 Apr 2023

PONE-D-22-29362R1 

Maternity and family leave experiences among female ophthalmologists in the United States 

Dear Dr. McGlumphy:

I'm pleased to inform you that your manuscript has been deemed suitable for publication in PLOS ONE. Congratulations! Your manuscript is now with our production department. 

Kind regards, 

on behalf of

Dr. Bidisha Banerjee 

Academic Editor

PLOS ONE